# Partial Ablation of Non-Myogenic Progenitor Cells as a Therapeutic Approach to Duchenne Muscular Dystrophy

**DOI:** 10.3390/biom11101519

**Published:** 2021-10-15

**Authors:** Zhanguo Gao, Aiping Lu, Alexes C. Daquinag, Yongmei Yu, Matthieu Huard, Chieh Tseng, Xueqin Gao, Johnny Huard, Mikhail G. Kolonin

**Affiliations:** 1Institute of Molecular Medicine, The University of Texas Health Science Center, Houston, TX 77030, USA; Zhanguo.Gao@uth.tmc.edu (Z.G.); Alexes.Daquinag@uth.tmc.edu (A.C.D.); Yongmei.Yu@uth.tmc.edu (Y.Y.); 2Center for Regenerative Sports Medicine, Steadman Philippon Research Institute, Vail, CO 81657, USA; alu@sprivail.org (A.L.); mhuard@sprivail.org (M.H.); xgao@sprivail.org (X.G.); 3M.D. Anderson Cancer Center, The University of Texas Health Science Center, Houston, TX 77030, USA; ctseng@mdanderson.org

**Keywords:** duchenne muscular dystrophy, platelet-derived growth factor receptor, adipose mesenchymal stromal cell

## Abstract

Duchenne muscular dystrophy (DMD), caused by the loss of dystrophin, remains incurable. Reduction in muscle regeneration with DMD is associated with the accumulation of fibroadipogenic progenitors (FAPs) differentiating into myofibroblasts and leading to a buildup of the collagenous tissue aggravating DMD pathogenesis. Mesenchymal stromal cells (MSCs) expressing platelet-derived growth factor receptors (PDGFRs) are activated in muscle during DMD progression and give rise to FAPs promoting DMD progression. Here, we hypothesized that muscle dysfunction in DMD could be delayed via genetic or pharmacologic depletion of MSC-derived FAPs. In this paper, we test this hypothesis in dystrophin-deficient *mdx* mice. To reduce fibro/adipose infiltration and potentiate muscle progenitor cells (MPCs), we used a model for inducible genetic ablation of proliferating MSCs via a suicide transgene, viral thymidine kinase (TK), expressed under the *Pdgfrb* promoter. We also tested if MSCs from fat tissue, the adipose stromal cells (ASCs), contribute to FAPs and could be targeted in DMD. Pharmacological ablation was performed with a hunter-killer peptide D-CAN targeting ASCs. MSC depletion with these approaches resulted in increased endurance, measured based on treadmill running, as well as grip strength, without significantly affecting fibrosis. Although more research is needed, our results suggest that depletion of pathogenic MSCs mitigates muscle damage and delays the loss of muscle function in mouse models of DMD.

## 1. Introduction

Duchenne muscular dystrophy (DMD) is the most common and severe form of muscular dystrophies caused by mutations in the dystrophin gene [1]. Progressive muscle weakness and degeneration due to the lack of dystrophin eventually leads to the loss of independent ambulation by the middle of the patient’s second decade and a fatal outcome due to cardiac or respiratory failure by the third decade [2]. In DMD patients, the loss of the sarcolemmal dystrophin complex reduces muscle membrane integrity and, consequently, triggers muscle fiber damage during muscle contraction [3,4,5,6,7,8]. This process results in an efflux of creatine kinase, an influx of calcium ions, and the recruitment of T cells, macrophages, and mast cells to the damaged muscle, contributing to progressive myofiber necrosis [8]. Importantly, DMD pathogenesis also involves fibrosis and deposition of fatty infiltrates (FI) in degenerating muscles. While several treatments have been approved for DMD, there is still no cure. 

Myogenic progenitor cells (MPCs), also known as Pax7+ satellite cells, lose their proliferation and differentiation capacities with age [9]. MPCs become dysfunctional (reduced proliferation and differentiation capacities) as DMD progresses, coincidentally with reduced muscle regeneration, aggravating fatty infiltration, and fibrotic tissue accumulation in skeletal muscle [10,11,12,13]. Coincidentally with MPC depletion and muscle pathology development, muscle fibrosis and FI are observed in both patients and animal models [14]. While fibrotic lesions and FI are secondary to sarcopenia and steroid therapy [15,16,17], there is a building body of evidence that they contribute to DMD pathogenesis [18,19,20]. There are also skeletal abnormalities observed in DMD patients, and heterotopic ossification (HO) is prominent in animal models of DMD [21]. Fibrosis, adipogenesis, and osteogenesis result from the differentiation of mesenchymal stromal cells (MSCs) that can differentiate into fibroblasts, adipocytes, and osteoblasts [22]. As with the bone marrow and other organs, MSCs in muscle are perivascular cells marked by the expression of platelet-derived growth factor receptors (PDGFRs) [23]. While myofibroblasts are derived from various progenitors, including MPC [24], MSC derivatives are the main source of pro-fibrotic extracellular matrix (ECM) proteins, such as collagen I, as well as of various cytokines modulating the immune tissue milieu [25]. It was demonstrated that MSCs could promote muscle regeneration after acute injury [26]. However, accumulating evidence indicates that in chronic disease, activated MSCs proliferate at the affected site and contribute to the population of fibroadipogenic progenitors (FAPs) fueling DMD [27,28]. There is accumulated evidence that during disease progression MSC-derived FAPs contribute to FI and fibrosis [19,26,29,30,31,32,33]. In addition, studies by our group showed that MSCs activated in pathology can directly depotentiate MPCs [34].

White adipose tissue (WAT) is an important reservoir of MSCs termed adipose stromal cells (ASCs), which serve as adipocyte progenitors [35]. The population of ASCs expands in obesity [36], the condition that is linked with DMD pathology [37]. In disease, ASCs can become mobilized and recruited to the sites of inflammation [38,39,40,41]. This has been demonstrated for tumors, in which ASCs become cancer-associated fibroblasts [36,42,43]. As with MSCs in other organs, ASCs are a key source of ECM and cytokines, which contribute to carcinoma progression [44]. We hypothesized that ASCs are recruited to muscle as it undergoes damage during DMD onset and contributes to FAPs. We rationalized that FAPs derived from muscle-resident MSCs and ASCs, exacerbating the disease progression, could be interrogated as a potential drug target. Here, we tested whether DMD pathogenesis can be delayed via genetic and pharmacologic depletion of MSC-derived and ASC-derived FAPs in the mouse model (Figure 1).

## 2. Materials and Methods

### 2.1. Animal Experiments

Mice were housed in the animal facility with a 12 h light/dark cycle at a temperature of 22–24 °C with free access to water and diet. Both strains, *Pdgfrb-TK* [45] and *mdx* [46], were in C57BL/6 background. For *Pdgfrb-TK* cell depletion, mice were injected (i.p.) daily with 50 mg/kg or 100 mg/kg body weight of ganciclovir (Sigma, Cat. # G2536, stock diluted with saline to 10 mg/mL) for 10 days. D-CAN peptide [47] was synthesized from D-amino acids, cysteine cyclized, and acetate salt chromatographically purified to 99% and quality-controlled (mass spectroscopy) by Ambiopharm. D-CAN (15 mg/kg) was administered by subcutaneous 12 semi-daily injections, as described [47,48]. Mice injected with PBS were used as a baseline control for GCV and D-CAN studies. Age-matched male littermates were used. An Exer-3/6 Treadmill 1055-SDRMAI-D60 equipped with Shock detection/auto-calibration was used. Mice were run at 12.5 m/s for exhaustion or at 9.8 m/s to measure fatigue resistance. Work performed was calculated as mouse weight (kg) × speed (m/min) × time (min) × incline (degree) × 9.8 m/s squared. A grip strength meter with a single sensor for mice along with standard pull bars and software (1027SM, Columbus Instruments, Columbus, OH, USA) was used to measure grip strength. 

### 2.2. Tissue Analysis

Frozen tissue sections were analyzed, as previously described [38,45,49,50]. H & E and Masson’s trichrome (Sigma, HT15-1KT) staining was imaged with Nikon Eclipse NiE microscope and quantified with Image J software. For immunofluorescence (IF), paraffin sections were blocked and the following primary antibodies were added in PBS with 0.05% Tween 20 overnight at 4 °C: anti-CD31 (BD Pharmingen, Franklin Lakes, NJ, USA, 553370, 1:300); anti-PDGFRβ (Abcam ab32570, 1:100); anti-CD206 (R & D Systems, Minneapolis, MN, USA, AF2534, 1:75); anti-CD68 (Invitrogen, Waltham, MA, USA, MA5-13324, 1:75). Upon washing, secondary donkey anti-goat Alexa488-IgG (Invitrogen, A11055, 1:200), anti-mouse Cy3-IgG (Jackson ImmunoResearch, West Grove, PA, USA, 715-165-150, 1:200) and anti-rabbit Cy3-IgG (Jackson ImmunoResearch, 711-166-152, 1:200) were used at RT, 1 h. Isolectin B4 was from Vector (B-1205, 1:50), and Streptavidin Alexa 488 was from Life Technologies (S32354, 1:200, Carlsbad, CA, USA). Nuclei were stained with Hoechst 33,258 (Invitrogen, Cat. # H3569). IF images were acquired with Carl Zeiss upright Apotome Axio Imager Z1/ZEN2 Core Imaging software.

### 2.3. Gene Expression Analysis

Total RNA was extracted using the Trizol Reagent (Life Technologies, 15596018). Complementary DNAs were generated using a High-Capacity cDNA Reverse Transcription Kit (Applied Biosystems, Waltham, MA, USA, 4368814). PCR reactions were performed on a C1000 Touch thermal cycler (Bio-Rad) using Q-PCR Master Mix (Gendepot, Katy, TX, USA, Q5600-005). The expression of mouse genes was normalized to 18S RNA. The Sybr green primers were as follows: 18S RNA: 5′-AAGTCCCTGCCCTTTGTACACA-3′, 5′-GATCCGAGGGCCTCACTAAAC-3′; Tgfb1 5′-CTCCACCTGCAAGACCAT-3′, 5′-CTTAGTTTGGACAGGATCTGG-3′; Pdgfrb, 5′-TGCCTCAGCCAAATGTCACC-3′ and 5′-TGCTCACCACCTCGTATTCC-3′.

### 2.4. Statistical Analysis

All statistical analyses were performed with GraphPad Prism 6 software. Experimental results are shown as mean +/− SEM. Two-tailed unpaired Student’s t-tests were performed on littermates unless otherwise indicated. *p* < 0.05 was considered significant. 

Data Sharing Statement: additional data and critical resources supporting the reported findings, methods, and conclusions will be available upon request.

## 3. Results

### 3.1. Genetic Depletion of PDGFRβ-Lineage Cells

For genetic ablation of PDGFRβ^+^ lineage MSCs, we used *Pdgfrb-TK* mice in which viral TK is expressed under the control of the PDGFRβ promoter [51]. In response to ganciclovir (GCV) treatment, only proliferating (but not quiescent) PDGFRβ-expressing cells are depleted in these mice. The proliferation of PDGFR+ FAPs is a requisite of both fibrosis and adipocyte expansion [52] demonstrated in DMD models [34] and models of aging [53]. Therefore, we predicted the inducible TK/GCV approach to be safe for healthy muscle because MSCs do not normally proliferate. An exception is WAT, in which PDGFRβ ASC depletion with the TK suicide gene prevents obesity development [45]. First, we confirmed that the approach is effective for depleting MSCs from skeletal muscle. *Pdgfrb-TK* mice were GCV-treated, and the gastrocnemius (GM) muscle was harvested, sectioned, and immunostained with antibodies against CD31 and PDGFRβ. Immunofluorescence quantification demonstrated that the frequency of PDGFRβ-expressing cells was significantly reduced after *Pdgfrb-TK* mice treated with GCV compared to non-treated control, while CD31 expression was not significantly different between treated and non-treated groups (Figure 2A,B).

We also analyzed the effect of the suicide gene therapy targeting proliferating MSC in *mdx* (dystrophin-null) mice, the conventional DMD model. We crossed *mdx* mice with *Pdgfrb-TK* mice and analyzed tissues in *mdx/Pdgfrb-TK* male mice injected with GCV. As above, we used *mdx*/*Pdgfrb-TK* male mice sham-injected with PBS as negative controls. Immunofluorescence revealed a clear depletion of PDGFRβ^+^ cells in GM, diaphragm, as well as cardiac muscle. Quantification of *Pdgfrb* expression (Figure 2D) is consistent with this observation. Isolectin B4-positive endothelial network remained unaffected in treated mice, consistent with the expected lack of treatment effect on quiescent PDGFRβ^+^ pericytes. To assess the implications of MSC depletion in dystrophic muscle, we measured the expression of *Tgfb1*, a major pro-fibrotic factor. In both muscle types measured, upon GCV treatment, *Tgfb1* significantly decreased (Figure 2E). While MSC depletion may partly account for changes in *Tgfb1* expression, macrophages are known to be a source of *Tgfb1* and aggravate DMD [54]. Thus, we analyzed infiltration of pro-inflammatory (CD68+CD206−) and alternatively-polarized (CD206+) macrophages. In both muscle types measured, upon GCV treatment, pro-inflammatory macrophage infiltration decreased, while alternatively-polarized macrophages remained apparent (Figure 2F. These changes in muscle stroma composition are consistent with reduced *Tgfb*1 expression.

### 3.2. Genetic Depletion of PDGFRβ-Lineage Cells Improves Muscle Function in Mdx Mice

Next, we analyzed DMD progression in *mdx*/*Pdgfrb-TK* mice challenged by repeated exhaustive treadmill running to expedite muscle degeneration and fibrosis. In order to assess changes upon treatment with GCV, we used *mdx*/*Pdgfrb-TK* male mice sham-injected with PBS, and *mdx* male mice negative for *Pdgfrb-TK* treated with GCV as negative controls. In the first experiment, we used 3-month-old mice that were challenged by three days of 10 min treadmill running to exhaustion immediately pre-treatment and 1 week post-treatment. Changes in endurance were assessed by comparing treadmill running time prior to treatment and 2 months after treatment completion (Figure 3A). Compared to wild-type (WT) male mice, used as a positive control, all *mdx* mice had a lower endurance, which has progressively decreased with age (Figure 3B). However, while control *mdx* mice initially tended to have higher endurance than GCV-treated mice, at the terminal time point (5 months old), GCV-treated mice had relatively higher endurance (Figure 3A,B). We also compared skeletal muscle performance in these mice by using a grip strength meter one month after treatment completion. As expected, WT mice had the highest grip strength. Notably, grip strength was higher in GCV-treated *mdx/Pdgfrb-TK* mice than in control *mdx* mice post-treatment (Figure 3C). 

We repeated this experiment in younger mice by using a different timing of exhaustion and GCV administration (Figure 3D). Mice at 1 month of age were challenged by 3 days of 10 min treadmill running to exhaustion, treated with GCV at 2 months of age, and then again challenged with two cycles of 3 days of 10 min treadmill running to exhaustion over the next month. Changes in endurance were assessed by comparing treadmill running time prior to treatment and 2 months after treatment completion (Figure 3D). Over this period, both sham-treated *mdx/Pdgfrb-TK* and GCV-treated *mdx* control progressively lost endurance (Figure 3D). In contrast, one of the GCV-treated *mdx/Pdgfrb-TK* mice increased treadmill running endurance and remained unchanged for the other mouse (Figure 3D).

At the terminal time point (5 months old), tissues were recovered from experimental mice to make frozen sections of GM, diaphragm, and cardiac muscle (Figure 3E and Appendix A). Staining with Masson’s trichrome was performed to assess myofiber morphology and measure collagen deposition. Compared to sham-treated *mdx/Pdgfrb-TK* and GCV-treated *mdx* negative controls, GCV-treated *mdx/Pdgfrb-TK* mice appeared to have a more normal myofiber organization and fewer necrotic lesions and acellular areas (Figure 3e). This was particularly apparent for the diaphragm stained with hematoxylin/eosin (Appendix A). However, quantification of Masson’s trichrome staining did not reveal a consistent difference in the overall amount of collagen in the three muscle types between GCV-treated *mdx/Pdgfrb-TK* mice and negative *mdx* controls (Figure 3F), suggesting that the effect is achieved only in some parts of the tissue. As expected, trichrome staining was different for mice from the two distinct GCV experiments, as mice were collected at different ages. Notably, in older *mdx/Pdgfrb-TK* mice treated with GCV, the diaphragm had significantly less collagen than in PBS-treated controls (data not shown). This suggests that the timing of FAP ablation in *mdx* determines the effect on fibrosis development.

Heterotopic ossification was also analyzed in experimental animals. Micro-CT overview of lumbar spine structure showed no difference between GCV- and PBS-treated *mdx/Pdgfrb-TK* mice. There was an obvious heterotopic bone formation in the spine surrounding muscles, which was comparable for GCV- and PBS-treated mice (Appendix A). (Appendix A) Micro-CT morphology of spine L5 and proximal tibia were similar for GCV- and PBS-treated mice (Appendix A), and there was no statistical difference for bone volume/tissue volume (BV/TV), an established histomorphometric porosity parameter (Appendix A). Quantification of microCT bone parameters indicated no significant difference for BV/TV, trabecular number (Tb.N), trabecular thickness (Tb.Th), and trabecular separation (Tb.Sp) between GCV- and PBS-treated dystrophic mice. However, L5 and proximal tibia BV density were significantly higher (*p* = 0.0302) in GCV-treated than for PBS-treated *mdx/Pdgfrb-TK* mice (Appendix A). While this improvement is modest, it suggests that improvement in muscle function resulting from FAP depletion may have indirect positive effects on other organs in DMD.

### 3.3. Pharmacological Depletion of Adipose Stromal Cells Improves Muscle Function in Mdx Mice

In the other approach, fat tissue-derived FAP ablation was performed pharmacologically. For this, we used a hunter–killer peptide targeting ASCs, which was developed and validated by our group in a series of publications [38,44,55]. This bi-modal molecule, termed D-CAN, is based on a cyclic peptide WAT7 (sequence CSWKYWFGEC) binding to ΔDCN, a proteolytic isoform of Decorin [49], which is displayed on the surface of *Pdgfrb*+ ASC but not endothelial or smooth muscle cells [56]. To enable ASC-directed cytotoxicity, WAT7 is linked to an amphipathic sequence KFAKFAKKFAKFAK, which, upon receptor-mediated cell internalization, induces mitochondrial membrane depolarization and apoptosis [57]. Because D-CAN selectively depleted *Pdgfrb*+ ASCs [47], we poised that it should reduce the bioavailability of FAPs. In a pilot experiment, 3 months-old *mdx* mice were injected with D-CAN or PBS (sham control) for 3 weeks, and endurance was analyzed by treadmill running immediately after treatment completion (Figure 4A). Running time was notably higher upon D-CAN treatment, indicating that it increases fatigue resistance (Figure 4B). In an independent experiment, 3 month-old *mdx* mice were challenged by three days of 10 min treadmill running to exhaustion prior to D-CAN treatment. After this conditioning, endurance was measured (Timepoint 1). Upon D-CAN treatment completion, mice were subjected to another cycle of three days of 10 min treadmill running to exhaustion. Then endurance was measured again (Timepoint 2). All control mice displayed a progressive decrease in endurance after PBS treatment (Figure 4B). In contrast, after D-CAN treatment, there was a significant increase in endurance in two out of three mice (Figure 4B). Combined analysis of work performed by all mice from each group demonstrates the effect of D-CAN (Figure 4B). At the terminal time point (5 months old), tissues were recovered from experimental mice to make frozen sections of GM, diaphragm, and cardiac muscle (Figure 4c and Appendix A). Hematoxylin/eosin (not shown) and Masson’s trichrome staining revealed that, compared to sham-treated controls, D-CAN-treated mice had better myofiber integrity and organization, as well as fewer necrotic lesions and acellular areas in GM muscle (Figure 4C). However, quantification of Masson’s trichrome staining did not reveal a consistent difference in the overall amount of collagen in the three muscle types between D-CAN-treated and negative control *mdx* mice (Figure 4D) suggesting that the effect is achieved only in some parts of the tissue. These observations, consistent with data from the proliferating MSC depletion with TK/GCV, suggest that muscle performance retention is, at least in part, independent of treatment effect on fibrosis. 

## 4. Discussion

Despite the progress in the understanding of DMD mechanisms and pathophysiology, it remains incurable. The standard of care is therapies that merely delay the disease progression [2]. Steroids, such as prednisone, which suppress inflammation and delay stem cell depletion [9], can slow down muscle damage and children retain the ability to walk by several more years. An oral corticosteroid, deflazacort (Emflaza), has been approved based on observations of prolonged muscle strength in clinical trials [2]. However, corticosteroid treatments have known side effects, including immune and metabolic [58,59].

Other approaches to DMD therapy have focused on restoring dystrophin to alleviate muscle weakness, which is problematic due to delivery constraints and the non-proliferative nature of myocytes. Eteplirsen (Exondys 51), a morpholino antisense oligomer that triggers excision of exon 51 during pre-mRNA splicing of the dystrophin RNA transcript, was designed to help individuals with a specific dystrophin mutation implicated in 13% of DMD cases [60]. Although it increases dystrophin production, no real improvement in muscle function has been reported to date. Emerging stem cell/dystrophin gene therapy and correction approaches based on CRISPR/Cas9 are evolving. Although they are effective in mouse models, their immediate value to patients is questionable due to efficacy and safety concerns [61,62,63,64,65,66]. New approaches to suppress DMD progression are required. Studies in animal models, showing that removal of a certain cell population, such as senescent cells, can impede DMD progression [67], have provided a possible alternative.

Accumulating evidence indicates that FAPs underlie DMD pathogenesis [68,69,70,71,72]. Here, we used genetic and pharmacological approaches to test FAPs as a potential DMD therapy target in *mdx* mice. We provide evidence that depletion of proliferating PDGFRβ+ FAP progenitors with TK/GCV suicide gene therapy, sparing quiescent PDGFRβ^+^ pericytes and endothelium, suppresses muscle damage and loss in endurance. The beneficial effects of PDGFRβ+ FAP depletion were linked with decreased infiltration of pro-inflammatory macrophages and reduced expression of a pro-fibrotic factor *Tgfb1*. In an independent approach, we show that depletion of adipose-derived FAP progenitors similarly delays the loss in muscle function. The concordant results from the two independent approaches suggest that the targeted subsets of MSC derivatives that partly, but likely not completely, overlap. Our combined results indicate that ablation of FAPs impedes muscle deterioration, although it does not have a dramatic positive effect on the skeletal pathology accompanying DMD progression. The effects were modest, likely due to transient and incomplete depletion of FAPs, which may re-populate the tissue from either remaining *Pdgfrb*+ MSCs or alternative progenitors. Subsequent studies with modified treatment regimens and timing of treatment will be required to characterize the effect of FAP progenitor depletion in detail. Our analysis suggests that the effect of FAP depletion on muscle function is only in part due to reduced ECM deposition. Other possible mechanisms, such as immunomodulation and MPC potentiation resulting from FAP depletion, likely explain the positive effects of treatments on muscle function retention in this study.

The translational considerations of the research carried out in an animal model remain to be determined. While the experimental drug D-CAN has shown promise in animal models, it is yet to enter clinical trials. Because its cellular target is known [49,56,57], other compounds direct it could potentially be designed to enable ASC-directed pharmacological therapy of DMD. This approach could be combined with drugs directed at other DMD-implicated targets, such as TGFβ and TNFα.

## 5. Conclusions

Our data show that MSCs, including ASCs, recruited to damaged muscle in DMD exacerbate the disease progression. We conclude that depletion of proliferating MSC and ASC suppresses muscle deterioration in the mouse DMD model in part through fibrosis-independent mechanisms, which remain to be established. Subsequent experiments with different D-CAN treatment doses and timing regimens will be needed to refine the therapeutic efficacy. Although these results are preliminary, they prime future studies in dystrophin/utrophin knockout (dKO) mice and larger animal models, such as the dystrophic dog (GRMD) and pig models that more closely mimic the DMD pathology. Improved pharmacological targeting of stromal cells may enable novel and more effective clinical approaches to delay DMD pathogenesis. We predict that pathogenic MSC ablation will synergize with other DMD treatment approaches. Information obtained from this study will help develop new therapeutic avenues for treating DMD.

## Figures and Tables

**Figure 1 biomolecules-11-01519-f001:**
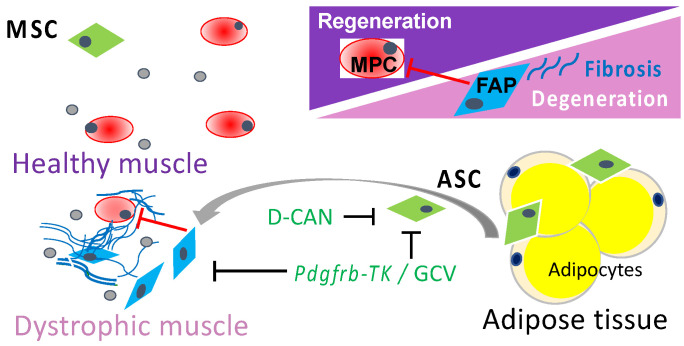
A schematic representation of the working hypothesis. In DMD muscle, resident MSCs and adipose stromal cells (ASCs) serve as a source of fibroadipogenic progenitors (FAPs) that promote fibrosis, suppress the activity of muscle progenitor cells (MPCs), and promote DMD pathogenesis. Ablation of proliferating MSCs (by TK/GCV suicide gene therapy) or ASCs (with D-CAN) is expected to suppress DMD progression.

**Figure 2 biomolecules-11-01519-f002:**
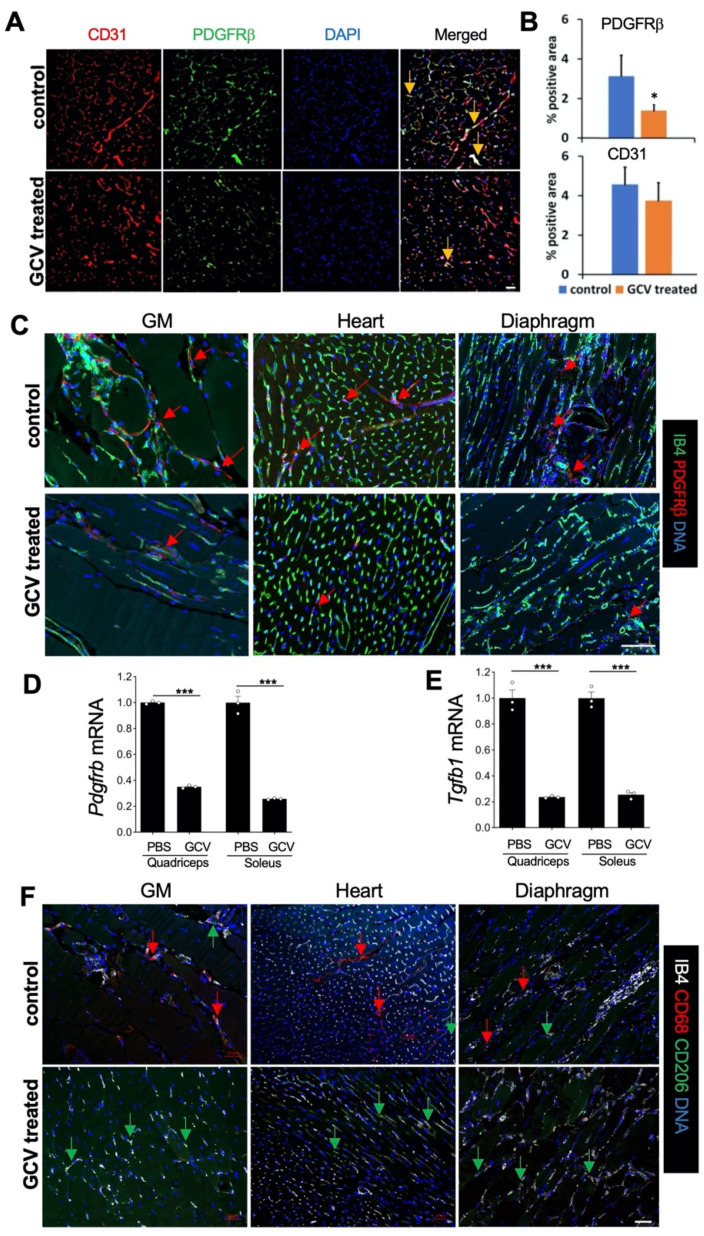
Ablation of *Pdgfrb*+ lineage MSCs in dystrophic muscle. (**A**) Representative IF (from *n* = 5) on gastrocnemius (GM) of *Pdgfrb-TK* mice showing that GCV treatment depletes PDGFRβ+ cells associated with CD31+ endothelium. (**B**) Data from (**A**) quantified based on the analysis of *n* = 10 view fields. (**C**) Representative IF on indicated muscle tissues from *mdx/Pdgfrb-TK* mice showing that GCV treatment ablates PDGFRβ+ cells (red arrows) associated with the endothelium stained with isolectin B4 (IB4). (**D**) RT-PCR showing the effect of *Pdgfrb*+ lineage cell ablation on the relative expression of *Pdgfrb* (normalized to *18S RNA*), *n* = 3. (**E**) RT-PCR showing the effect of *Pdgfrb*+ lineage cell ablation on the relative expression of *Tgfb1* (normalized to *18S RNA*) *n* = 3. (**F**). Representative IF on indicated muscle tissues from *mdx/Pdgfrb-TK* mice showing that GCV treatment reduces the frequency of CD68+ macrophages (red arrows) but not CD206+ macrophages (green arrows). * *p* < 0.05; *** *p* < 0.01 (Student’s *t*-test). Scale bar: 50 µm.

**Figure 3 biomolecules-11-01519-f003:**
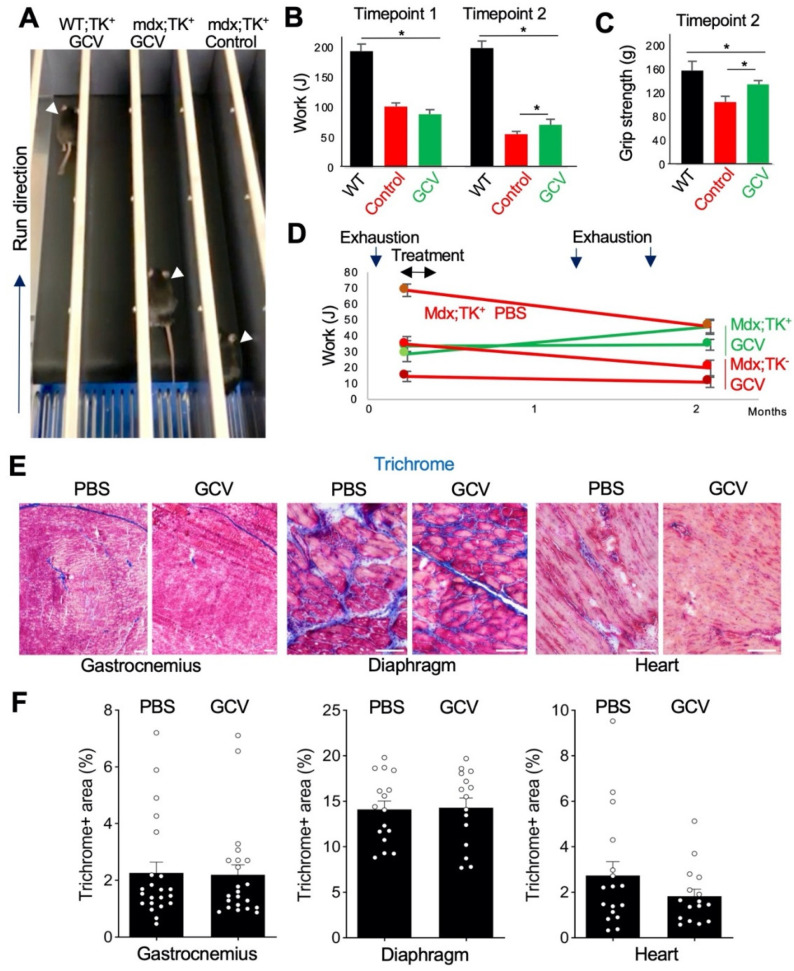
Depletion of PDGFRβ-lineage cells in *mdx* mice. (**A**) A snapshot of the treadmill test revealing increased endurance *of mdx/Pdgfrb-TK* mice treated with GCV compared to control (PBS) treatment. (**B**) Long-term endurance improvement in *mdx*/*PDGFRb-TK* mice treated with GCV. *****
*p* < 0.05, *n* = 4 (Student’s *t*-test). (**C**) Long-term grip strength improvement in *mdx/PDGFRb-TK* mice treated with GCV. *****
*p* < 0.05, *n* = 4 (Student’s t-test). (**D**) Timeline of experiment showing endurance improvement in *mdx/Pdgfrb-TK* mice treated with GCV. (**E**) Masson’s trichrome staining showing decreased regression and disorganization of myofibers and reduced fibrosis in muscle of *mdx/Pdgfrb-TK* mice treated with GCV. (**F**) Quantification of Masson’s trichrome staining-positive area in gastrocnemius, diaphragm, and cardiac muscle, *n* > 15. Scale bar: 50 µm.

**Figure 4 biomolecules-11-01519-f004:**
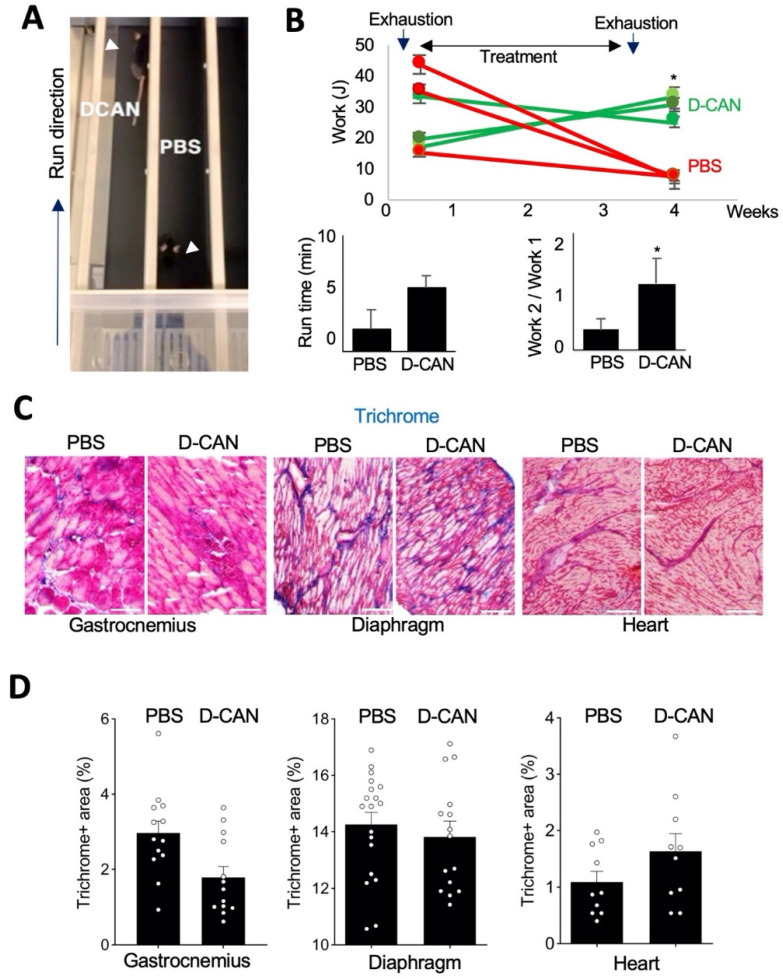
(**A**) A snapshot of the treadmill test revealing increased endurance of *mdx* mice treated with D-CAN. (**B**) The timeline of experiment showing endurance improvement in *mdx* mice treated with D-CAN graphed underneath. Work at timepoint 1 was set as 1, and relative change at timepoint 2 is shown. *N* = 3. * *p* < 0.05. (**C**) Masson’s trichrome staining of gastrocnemius muscle from individual mice showing less regression and disorganization of fibers in D-CAN-treated mice. (**D**) Quantification of Masson’s trichrome staining-positive area in gastrocnemius, diaphragm, and cardiac muscle, *N* > 12. Scale bar: 50 µm.

## Data Availability

The data presented in this study are available on request from the corresponding author.

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
