# Peer review of "Partial Ablation of Non-Myogenic Progenitor Cells as a Therapeutic Approach to Duchenne Muscular Dystrophy"

_biomolecules, 2021, doi:10.3390/biom11101519_

Round 1

Reviewer 1 Report

Journal: Biomolecules

ID: biomolecules-1305363

Title: Ablation of Non-Myogenic Progenitor Cells as a Therapeutic Approach to Duchenne Muscular Dystrophy

In this manuscript, the authors concentrate on: fibroadipogenic progenitors (FAPs), myofibroblast transdifferentiation, genetic and pharmacological inhibition of FAPs, and dystrophic fibrosis.

Although very interesting, this study shows some limitations, in order of importance (high to minor):

HIGH

  • Although difficult to obtain this research would be improved if samples from DMD patients’ muscle biopsies and/or cardiomyocytes (and indeed other DMD disease-relevant cell types derived from DMD patients’ iPSCs e.g. cardiac fibroblasts, macrophages, ECs; see DOI: 10.1016/j.stem.2020.05.004).
  • TGF-β1 quantification should be performed.
  • The frequency of Ly6C+ cells in skeletal and cardiac tissue should be determined.
  • Tissue from more than one muscle should be analysed for Fig. 1. Additionally, more methodological detail should be given regarding the specific area of cardiac tissue used for quantification purposes. Low magnification images showing the whole cross-section of the hearts should be shown.

MEDIUM

  • Impaired FAP clearance by infiltrating monocytes is not mentioned, also macrophage-mediated TNF-induced FAP apoptosis should be discussed. See: Lemos, D., Babaeijandaghi, F., Low, M. et al. Nilotinib reduces muscle fibrosis in chronic muscle injury by promoting TNF-mediated apoptosis of fibro/adipogenic progenitors. Nat Med 21, 786–794 (2015). https://doi.org/10.1038/nm.3869.
  • 4 lines 161-164, regarding the receptor that is bound by the amphipathic D-CAN sequence quoted in this manuscript, I presume this is also decorin which is expressed not only by ASCs but also by vascular endothelial cells and smooth muscle cells. Can the authors discuss this and if there are associated cell-specificity concerns.
  • Although the authors hint at this point in the conclusion section, they should include a few more lines that discuss the translational considerations of the research e.g. this is a study carried out in an animal model. Also, the drug target proposed here, has there been clinical trials? Please present more information on targeting this pathway from a clinical perspective e.g. is there a preference between gene and pharmacological therapy? Lastly, the authors should add that this approach could be combined with other drugs that target other pathomechanisms of DMD e.g. combination of D-CAN with TGF or TNF modulation.
  • The ages of animals used are not clear at certain points of the manuscript e.g. methods what age were the mice when treated with ganciclovir and D-CAN; text p. 4 line 134 “At the terminal time point,…” rather confusing since in the previous paragraphs two different experimental setups are presented, which time point (presumably age or possible treatment) are the authors referring to here?; ages of the animals are not clearly understandable from the figures or the figure legends.
  • References that should be included and discussed in the manuscript:

    • DOI: 10.33594/000000196
    • https://doi.org/10.1038/s41467-019-10438-z
    • DOI: 10.26508/lsa.202000660
    • 1172/JCI133303
    • http://www.genesdev.org/cgi/doi/10.1101/gad.234468.113

MINOR

  • The title should be changed to reflect that FAPs are not totally ablated by either genetic or pharmacological approaches.
  • 4 line 133 “while the other one has maintained it” revise language to more formal style; line 142; line 143 the diaphragm, treated not “treaded”.
  • 5 lines 167-171 please revise language usage here e.g. “Then endurance….” this sounds too colloquial, measured is used repetitively.
  • 7 line 213 “etiology” this reviewer would remove this; line 214 insert ‘that’ before “merely”; line 217 the sentence beginning “However hormonal …..” is disjointed from the thread of the paragraph's argument, please delete or clarify what the point is here.
  • 8 line 241 give examples of fibrosis-independent mechanisms; lines 226-227 revise grammar, suppress rather than suppresses, ‘loss of endurance’ not “the loss in endurance”; line 244 refers to the GRMD model the authors should also include the MD pig model, DMDF? Typo
  • Please use italics for mdx.
  • Errant extra spaces are present in multiple locations throughout the test please carefully check the spacing.

Author Response

We are grateful to the reviewers for identifying the opportunities to strengthen the work. We addressed all the points raised by performing new experiments as discussed below. Figure 2 contains new display items. New Figure S1 was added and old Figure S1 is renamed S2.

In this manuscript, the authors concentrate on: fibroadipogenic progenitors (FAPs), myofibroblast trans-differentiation, genetic and pharmacological inhibition of FAPs, and dystrophic fibrosis. Although very interesting, this study shows some limitations, in order of importance (high to minor):

Response: We thank the Referee for finding the study interesting and for the constructive suggestions to improve the manuscript, which we have followed.

HIGH

Although difficult to obtain this research would be improved if samples from DMD patients’ muscle biopsies and/or cardiomyocytes (and indeed other DMD disease-relevant cell types derived from DMD patients’ iPSCs e.g. cardiac fibroblasts, macrophages, ECs; see DOI: 10.1016/j.stem.2020.05.004).

Response: Indeed, iPSCs derived from patients have been used to establish ex vivo models of cultured human cells with dystrophin mutations causing DMD. They have potential value in establishing dystrophin function and screens for drugs improving myocyte survival and function. For our study, however, it is not clear how such models could be complementary, even if we had managed to identify collaborators willing to provide biopsies. The role of FAPs and, more importantly, the functional consequences of their depletion during DMD progression can only be studied in animal model. Therefore, the use of human cells is not within the scope of this study.

TGF-β1 quantification should be performed.

Response: Thank you for pointing out that expression of Tgfb1, a major pro-fibrotic factor, is an important variable. We have now measured Tgfb1 expression in experimental samples by RT-PCR. In both muscle types measured, upon GCV treatment, Tgfb1 significantly decreased. This is consistent with the improvement in muscle histology and reduced proinflammatory macrophage infiltration. Data are presented in new Figure 2D and discussed (pages 4 and 10).

The frequency of Ly6C+ cells in skeletal and cardiac tissue should be determined.

Response: The reviewer again brings a point that treatments could affect the numbers of pro-inflammatory (Ly6C+CD68+CD206-) macrophages. The use of Ly6C antibodies in tissue sections is prone to artifacts. Therefore, we have analyzed infiltration of pro-inflammatory (CD68+CD206-) and alternatively-polarized (CD206+) macrophages by using antibodies previously validated in our studies (doi: 10.1152/ajpendo.00314.2016). In both muscle types measured, upon GCV treatment, pro-inflammatory macrophage infiltration decreased, while alternatively-polarized macrophages remained abundant. This is consistent with the improvement in muscle histology and reduced Tgfb1 expression. Data are presented in new Figure  2F and discussed (pages 4 and 10).

Tissue from more than one muscle should be analyzed for Fig. 1. Additionally, more methodological detail should be given regarding the specific area of cardiac tissue used for quantification purposes. Low magnification images showing the whole cross-section of the hearts should be shown.

Response: Figure 1 is a schematic representation and does not contain muscle analysis. Because Figures 3, and 4 already contain analyses of three muscle types, we assume  that Figure 2 was refereed to. We now updated it with new data. New Figures 2C and 2F contain additional analysis of gastrocnemius, diaphragm and cardiac muscles.

Left ventricle heart section images are shown in Figures 3 and 4. We have now provided low magnification images showing the whole cross-section of the hearts in new Supplemental Figure S1A.

MEDIUM

Impaired FAP clearance by infiltrating monocytes is not mentioned, also macrophage-mediated TNF-induced FAP apoptosis should be discussed. See: Lemos, D., Babaeijandaghi, F., Low, M. et al. Nilotinib reduces muscle fibrosis in chronic muscle injury by promoting TNF-mediated apoptosis of fibro/adipogenic progenitors. Nat Med 21, 786–794 (2015). https://doi.org/10.1038/nm.3869.

Response: We agree that this is an important publication reporting results consistent with our findings, which is now cited and discussed (Page 4).

4 lines 161-164, regarding the receptor that is bound by the amphipathic D-CAN sequence quoted in this manuscript, I presume this is also decorin which is expressed not only by ASCs but also by vascular endothelial cells and smooth muscle cells. Can the authors discuss this and if there are associated cell-specificity concerns.

Response: We are now making clearer that the receptor of D-CAN is not full length Decorin (which is located in the ECM), but is rather its proteolytic isoform generated by MMP14 and displayed on ASC surface (New Ref DOI: 10.3390/cells9122646). We now discuss this for clarity in the revised paper (Page 5).

Although the authors hint at this point in the conclusion section, they should include a few more lines that discuss the translational considerations of the research e.g. this is a study carried out in an animal model. Also, the drug target proposed here, has there been clinical trials? Please present more information on targeting this pathway from a clinical perspective e.g. is there a preference between gene and pharmacological therapy? Lastly, the authors should add that this approach could be combined with other drugs that target other pathomechanisms of DMD e.g. combination of D-CAN with TGF or TNF modulation.

Response: We are now expanded the Discussion along these lines (Page 10) as follows:

The translational considerations of the research carried out in an animal model remain to be determined. While the experimental drug D-CAN has shown promise in animal models, it is yet to enter clinical trials. Because its cellular target is known [49, 56, 57] other compounds direct it could potentially be designed to enable ASC-directed pharmacological therapy of DMD. That this approach could be combined with drugs directed at other DMD-implicated targets, such as TGFβ and TNFa.”

The ages of animals used are not clear at certain points of the manuscript e.g. methods what age were the mice when treated with ganciclovir and D-CAN; text p. 4 line 134 “At the terminal time point,…” rather confusing since in the previous paragraphs two different experimental setups are presented, which time point (presumably age or possible treatment) are the authors referring to here?; ages of the animals are not clearly understandable from the figures or the figure legends.

Response: We have now clarified in the text that the terminal time point was 5 months of age for both GCV and D-CAN experiments.

References that should be included and discussed in the manuscript:

DOI: 10.33594/000000196 https://doi.org/10.1038/s41467-019-10438-z DOI: 10.26508/lsa.202000660

1172/JCI133303 http://www.genesdev.org/cgi/doi/10.1101/gad.234468.113

Response: Thank you for the suggestion. We have now cited these publications in the revised discussion (page 10).

MINOR

The title should be changed to reflect that FAPs are not totally ablated by either genetic or pharmacological approaches.

Response: Subject to editorial discretion, we have changed the title to “Partial Ablation of Non-Myogenic Progenitor Cells as a Therapeutic Approach to Duchenne Muscular Dystrophy”

4 line 133 “while the other one has maintained it” revise language to more formal style; line 142; line 143 the diaphragm, treated not “treaded”.

5 lines 167-171 please revise language usage here e.g. “Then endurance….” this sounds too colloquial, measured is used repetitively.

7 line 213 “etiology” this reviewer would remove this; line 214 insert ‘that’ before “merely”; line 217 the sentence beginning “However hormonal …..” is disjointed from the thread of the paragraph's argument, please delete or clarify what the point is here.

8 line 241 give examples of fibrosis-independent mechanisms; lines 226-227 revise grammar, suppress rather than suppresses, ‘loss of endurance’ not “the loss in endurance”; line 244 refers to the GRMD model the authors should also include the MD pig model, DMDF? Typo

Please use italics for mdx.

Errant extra spaces are present in multiple locations throughout the test please carefully check the spacing.

Response: Thank you. All these text issues have been fixed, in the revised paper, as follows:

“while the other one has maintained it” changed toand remained unchanged for the other mouse”

Then endurance” changed to  “After this conditioning, endurance was measured”

“etiology” replaced with “mechanisms and pathophysiology”

“hormonal” replaced with “corticosteroid”

“fibrosis-independent mechanisms”: we are referring to a conclusion of this study.

changed to  “fibrosis-independent mechanisms, which remain to be established”

“depletion .. suppresses” is correct grammar              

“loss in endurance” corrected 

Change: “dystrophic dog (GRMD) and pig models that more closely mimic the DMD pathology”

All typos indicated and extra spaces have been fixed. Gene names were made italic.

Reviewer 2 Report

This communication paper from Gao et al., builds on previous mouse models and methods developed by the authors and could be of interest for the skeletal muscle/muscular dystrophy community. While targeting FAPs to treat DMD is not a new idea, the authors repurpose ganciclovir (GVC) or D-CAN treatment to deplete senescent cells; if proven correct this approach can be easily translatable to humans.

This communication paper could also improve our understanding of how senescence is associated with DMD. 

However, the reviewer thinks that the authors should improve the manuscript before final acceptance. 

General comments:

The authors assumed that their pharmacological treatments only depleted PDGFRbeta+ FAPs cells, but there is no evidence supporting this statement in their current manuscript. GVC (and I guess D-CAN works similarly) can target all senescent cells, and unlike the PDGFRalpha+ canonical FAP marker, PDGFRbeta is also expressed by other cells types including satellite cells, pericytes, and endothelial cells. Here, the authors only explored the impact on endothelial CD31+ cells. 

The reviewer suggests to further investigate and discuss how the authors’ pharmacological approaches impact senescence; authors could refer to recently published papers, such as Sugihara, et al. Sci Rep (2020) https://doi.org/10.1038/s41598-020-73315-6

The tissue histology figures (i.e., showing CD31 and PDGFRbeta IF, H&E, and Masson’s trichome staining) could be of highermagnification. It would enable readers to visually assess the improvement of muscle visually histology (e.g., « a more normal myofiber organization and fewer necrotic lesions and acellular areas » or « less regression and disorganization of fibers »).

 Major comments: 

Figure 2. It is unclear for the reviewer how many mice have been used for the quantification. 

What do the yellow arrows point out? Are FAP supposed to be CD31+ PDGFRbeta+ or CD31- PDGRFbeta +?

Unlike the canonical PDGFRalpha marker, PDGFRbeta is also expressed by other cell types, including satellite cells and pericytes.Could the authors confirm that GVC treatment specifically targeted and removed FAPs and not other cell types?

 Figure 3 & 4. Would it be possible to visualize the individual data points instead of a bar graph showing mean values only?

 Minor comments:

Figure 2. The authors could include the WT in the CD31/PDGFRbeta IF and compare them to the control and GVC treatment as they did for Figure 3.

The authors could also perform the CD31 and PDGFRbeta IF on tissue after completing their running experiments to confirm FAP depletion and determine whether FAPs repopulate the dystrophic muscle overtime at the time of the experiment.

Author Response

We are grateful to the reviewers for identifying the opportunities to strengthen the work. We addressed all the points raised by performing new experiments as discussed below. Figure 2 contains new display items. New Figure S1 was added and old Figure S1 is renamed S2.

This communication paper from Gao et al., builds on previous mouse models and methods developed by the authors and could be of interest for the skeletal muscle/muscular dystrophy community. While targeting FAPs to treat DMD is not a new idea, the authors repurpose ganciclovir (GVC) or D-CAN treatment to deplete senescent cells; if proven correct this approach can be easily translatable to humans. This communication paper could also improve our understanding of how senescence is associated with DMD. However, the reviewer thinks that the authors should improve the manuscript before final acceptance.

Response: We thank the Referee for acknowledging the importance and translational relevance of the study and for the constructive suggestions to improve the manuscript, which we have followed.

General comments:

The authors assumed that their pharmacological treatments only depleted PDGFRbeta+ FAPs cells, but there is no evidence supporting this statement in their current manuscript. GVC (and I guess D-CAN works similarly) can target all senescent cells, and unlike the PDGFRalpha+ canonical FAP marker, PDGFRbeta is also expressed by other cells types including satellite cells, pericytes, and endothelial cells. Here, the authors only explored the impact on endothelial CD31+ cells. 

Response: As we now clarify in Results (page 5) and Discussion (page 10), PDGFRbeta-Cre-TK/GCV and D-CAN both target subsets of MSC derivatives that partly, but likely not completely, overlap. The former targets PDGFRb+ proliferating (rather than senescent) cells, and spares most pericytes (which are also PDGFRb+ but mainly not proliferating). The latter targets both PDGFRa+ and PDGFRb+ ASCs, but has a selectivity for PDGFRb+ cells. As now emphasized (e.g. pages 4 and 5), neither treatment targets satellite cells or endothelial CD31+ cells: neither express PDGFRb or DDCN, the D-CAN receptor. Please see new data added to Figure 2.

The reviewer suggests to further investigate and discuss how the authors’ pharmacological approaches impact senescence; authors could refer to recently published papers, such as Sugihara, et al. Sci Rep (2020) https://doi.org/10.1038/s41598-020-73315-6

Response: We now cite and discuss this important paper on page 10. Because in that study senescence was induced in myogenic progenitors (in addition to MSCs), the beneficial effect of senolytic treatment could be to a large extent due to removal of senescent myocytes.

The tissue histology figures (i.e., showing CD31 and PDGFRbeta IF, H&E, and Masson’s trichome staining) could be of higher magnification. It would enable readers to visually assess the improvement of muscle visually histology (e.g., « a more normal myofiber organization and fewer necrotic lesions and acellular areas » or « less regression and disorganization of fibers »).

Response: We have now included high magnification images of H/E stainings in new Supplementary Figure S1B.

Major comments: 

Figure 2. It is unclear for the reviewer how many mice have been used for the quantification. 

What do the yellow arrows point out? Are FAP supposed to be CD31+ PDGFRbeta+ or CD31- PDGRFbeta +?

Response: We apologize for the legend unclarity, which is now fixed. As now indicated in updated legend, 5 mice / cohort were analyzed and representative images are shown. FAPs are supposed to be PDGFRb+CD31-. PDGFRb+ cell do not express CD31. The arrows point to perivascular PDGRFb+ cells associated with CD31+ blood vessels. Yellow color is due to cell overalp, which is unavoidable at such resolution for epifluorescence. New Fig. 2C has high-magnification images showing that PDGRFb+ cells and endothelial cells are separate.

Unlike the canonical PDGFRalpha marker, PDGFRbeta is also expressed by other cell types, including satellite cells and pericytes. Could the authors confirm that GVC treatment specifically targeted and removed FAPs and not other cell types?

Response: As commented above, GCV in the TK model targets PDGFRb+ proliferating cells, and spares endothelial cells and most pericytes (which are mainly not proliferating). This is now emphasized in updated Results (page 5) and Discussion (page 10).

Figure 3 & 4. Would it be possible to visualize the individual data points instead of a bar graph showing mean values only?

Response: We have now re-plotted Figure 3F and Figure 4D as dot-plots.

Minor comments:

Figure 2. The authors could include the WT in the CD31/PDGFRbeta IF and compare them to the control and GVC treatment as they did for Figure 3. 

Response: Mice not treated with GCV are essentially WT. The reason we added WT mice in Figures 3 and 4 is to compare them to mdx mice, which were treated. This does not apply to Figure 2.

The authors could also perform the CD31 and PDGFRbeta IF on tissue after completing their running experiments to confirm FAP depletion and determine whether FAPs repopulate the dystrophic muscle overtime at the time of the experiment.

Response: We appreciate the suggestion. While we did not perform animal studies to explore FAP repopulation over time, we have now re-analyzed the available tissues to address the question. Data are presented in new Fig. 2C. It is evident that, while PDGFRb+ cells are depleted, CD31+ endothelial cells remain and, in fact, vasculature appears to be denser in treated mice, which is an indirect indication of muscle functionality. Quantification of Pdgfrb expression in new Fig 2D is consistent with this observation.

Round 2

Reviewer 1 Report

I find that the Authors' additions have made the manuscript stronger.

One last additional change I would recommend before publication is to add the N numbers or 'observation' N number to Fig. 2 D&E, Fig. 3 F, and Fig. 4 B&D.

I send my commendations to the Authors for their attentive review of their manuscript.   

Author Response

Thank you: we have added this information to the legends.